# Relationship between protein conformational stability and its immunogenicity when administering antigens to mice using adjuvants—Analysis employed the CH$_2$ domain in human antibodies

**Kosuke Oyama**, **Tadashi Ueda** *

Graduate School of Pharmaceutical Sciences, Kyushu University, Fukuoka, Japan

* ueda@phar.kyushu-u.ac.jp

**Data Availability Statement:** All relevant data are within the manuscript.

## Abstract

Antigen-presenting cells (APCs) play a crucial role in the immune system by breaking down antigens into peptide fragments that subsequently bind to major histocompatibility complex (MHC) molecules. Previous studies indicate that stable proteins can impede CD4$^+$ T cell stimulation by hindering antigen processing and presentation. Conversely, certain proteins require stabilization in order to activate the immune response. Several factors, including the characteristics of the protein and the utilization of different adjuvants in animal experiments, may contribute to this disparity. In this study, we investigated the impact of adjuvants on antigen administration in mice, specifically focusing on the stability of the CH$_2$ domain. Consequently, the CH$_2$ domain induced a stronger IgG response in comparison to the stabilized one when using Alum and PBS (without adjuvant). On the other hand, animal experiment using Freund's adjuvant showed the opposite results. These findings indicate the significance of considering the intrinsic conformational stability of a protein when eliciting its immunogenicity, particularly within the context of vaccine development.

## Introduction

Vaccines aim to stimulate the immune response, and in vaccine development, vaccine immunogenicity can be enhanced by modifications of protein antigens, such as amino acid mutation [1], chemical modification [2] or oligomerization [3]. The relationship between protein conformational stability and vaccine immunogenicity is emphasized [4, 5], as lower stability accelerates protein digestion [6–9]. Antigen-presenting cells (APCs) utilize proteases to degrade antigens into peptide fragments, which then bind to major histocompatibility complex (MHC) molecules. The peptide-MHC complexes are presented on the cell surface and recognized by T cells. This interaction activates T cells and initiates an immune response against the specific antigen [10, 11]. There is evidence suggesting that stable proteins have a lower ability to stimulate helper T cells [12–15], and/or antibody responses [12, 15–18], because of inefficient

**Funding:** Financial support for this study was granted through operational resources from Kyushu University under the auspices of the Ministry of Education, Culture, Sports, Science, and Technology (MEXT), as well as the "COVID-19 Drug and Vaccine Development Donation Account" Project, which received support from Sumitomo Mitsui Trust Bank. The study design, data collection, and analysis were supported by these funding sources. Furthermore, the data collection, analysis, and publication of the manuscript were supported by a grant from the Japan Agency for Medical Research and Development (Grant No. 223fa827004h0007) to TU.

**Competing interests:** The authors have declared that no competing interests exist.

antigen processing and presentation. On the other hand, certain proteins can only trigger an immune response when they are stabilized [19, 20]. The strategy to enhance the immune response to protein antigens by manipulating protein conformational stability in vaccine design is still not fully understood [4, 5]. Previous studies about relationship between protein stability and the immunogenicity were employed different proteins with different charges, molecular weights, intrinsic conformational stabilities and different adjuvants. We have recently shown that mice immunized with the SARS-CoV-2 spike protein mixed with Alum produced neutralizing antibodies in sera. Mice immunized with the protein mixed with CFA/IFA did not produce an effective neutralizing response, even though both adjuvants induced an IgG response [21]. Similar findings have been reported in the experiments involving MERS-CoV RBD-Fc complex [22]. Therefore, we focused on Alum and Freund's adjuvant that have been primarily utilized to administer antigens to mice in this study.

Adjuvants are essential for augmenting the effectiveness of vaccination procedures as they function as carriers, depots and stimulators of the immune response [23]. Classical studies have shown that the presence of CFA containing inactivated *Mycobacterium tuberculosis* can stimulate T-lymphocytes to develop a Th1 profile, leading to a robust delayed type of hyper-sensitivity response against autoantigens. Conversely, the absence of inactivated *Mycobacterium tuberculosis* tends to promote the differentiation of T-lymphocytes towards a Th2 profile, characterized by predominantly strong antibody production [24]. Furthermore, CFA triggered an upregulation in Toll-like receptor 2 expression [25], IFA influenced a reduction in the protein's conformational stability [26], and the discussion focused on the relationships between antigens and adjuvants, such as oil, and their impact on the effectiveness of vaccines [27]. On the contrary, although the mechanism of action of Alum was not fully understood at the time, it was also noted that alum triggers cell death, resulting in the release of host cell DNA. This DNA acts as a robust endogenous immunostimulatory signal that enhances the adjuvant effect of alum [28]. Furthermore, the impact of alum on the target cell at cellular and molecular levels, along with the resulting innate and adaptive responses, is crucial for the strategic development of efficacious vaccines for various diseases, as outlined in a study by Ghimir [29].

IgG is a dimeric glycoprotein with a conserved N-glycosylation site in each of its $CH_2$ domains in the effector region, referred to as Fc [30]. The $CH_2$ domain in human IgG1 is known for its inherent instability [31], a characteristic that may lead to the production of Anti-Drug Antibodies, as unstable proteins have a tendency to aggregate [32]. The process of aggregation is closely associated with their immunogenicity [33]. The conformational stability of the molecule can be increased by 20°C through the introduction of a disulfide bond (L242C/K334C, the residues are numbered according to EU numbering) [34]. The structural similarity between the $CH_2$ domain mutant and the intact $CH_2$ domain is indicated by the comparable Fc receptor binding ability of the Fc mutant to that of the intact Fc [31]. The thermally stabilized $CH_2$ domain mutant is referred to as mut20.

Herein, the study investigated the correlation between the conformational stability of a protein and its immunogenicity when exposed to two different adjuvants, Alum and Freund's adjuvant, which are most commonly used. This experiment employed an intact human $CH_2$ domain and mut20. The administration of the $CH_2$ domain and its stabilized variant to mice resulted in distinct immune responses, indicating interactions between the protein and mineral oil in Freund's adjuvants. On the other hand, Delamarre et al. showed that the administration of ribonuclease A (RNase A) with Alum in mice led to increased IgG production in comparison to the administration of ribonuclease S (RNase S) [19, 20]. Additionally, it was observed that RNase A exhibited greater stability than RNase S, in which RNase S is a variant of RNase A that has been cleaved between the amino acids Ala20 and Ser21. These results seem to conflict with the outcomes of our investigation on the $CH_2$ domain in animal

experiments involving Alum. Consequently, we investigated the peptide quantity through a cathepsin digestion assay to elucidate the underlying reasons for the disparate immune responses resulting from variations in protein stability between $CH_2$ domain and mut20, as well as between RNase A and RNase S, in order to explore the importance of conformational stability on the immune response when administering protein antigens to mice with Alum adjuvant by incorporating these proteins in our research.

## Materials and methods

### Materials and reagents

The human non-glycosylated $CH_2$ domain and mut20 from *Pichia pastoris* were prepared according to our previous study [35, 36]. Hen lysozyme and RNase A were purchased from Nacalai Tesque, Inc. (Japan). Cathepsin B and RNase S were purchased from Sigma Aldrich (USA). The reduced and carboxymethylated lysozyme was prepared according to a previous study [37]. All the other reagents and chemicals used in the study were of analytical grade or higher quality.

### Immunization of mice

All animal experiments were conducted according to the relevant national and international guidelines in the Act on Welfare and Management of Animals (Ministry of Environment of Japan) and the Regulation of Laboratory Animals (Kyushu University) and under the protocols approved by the Institutional Animal Care and Use Committee review panels at Kyushu University (Animal protocol number: A21-053-0).

Seven-week-old female BALB/c mice were injected intraperitoneally (i.p.) according to the immune schedule (Fig 1A). We opted for the i.p administration route based on a study by Delamarre et al., [19] which illustrated the efficacy of this particular route of administration. The administration route led to an augmentation in antibody production compared to intramuscular administration. Moreover, the study observed that the method of administration did not significantly impact the stability of proteins and their immunogenic characteristics. Briefly, the $CH_2$ domain or mut20 (30 μg/shot) was administered using either Alum adjuvant, CFA adjuvant or IFA adjuvant. The protein samples (100 μg/shot) were administered in PBS at 0 days. Booster immunizations were administered using the same protein at a dose of 30 μg/shot, dissolved in either Alum adjuvant or IFA, or with a dose of 100 μg/shot, dissolved in PBS on days 7 and 14. The successive administration adheres to a standardized protocol within Freund's adjuvant. Typically, Complete Freund's Adjuvant (CFA), renowned for its ability to induce strong inflammation, is commonly used only for the initial injection, followed by the administration of Incomplete Freund's Adjuvant (IFA) to enhance immune responses. The immunization schedule we propose, with 1-week intervals, may appear shorter when contrasted with the conventional method that typically utilizes a 2-week interval for protein vaccine immunization. The immunization protocol employed in this study was developed following methodologies previously established in our laboratory [17, 18, 21, 36, 38]. Blood samples were drawn every week under anesthesia using sevoflurane to remove pain. Serum antibodies were detected using ELISA. Following terminal blood collection, mice were sacrificed by cervical dislocation under anesthesia.

### ELISA

96-well ELISA plates were coated with $CH_2$ domain or mut20 in coating buffer (15 mM sodium carbonate, 34.9 mM sodium bicarbonate, 0.2% sodium azide, 1 L of distilled water, pH

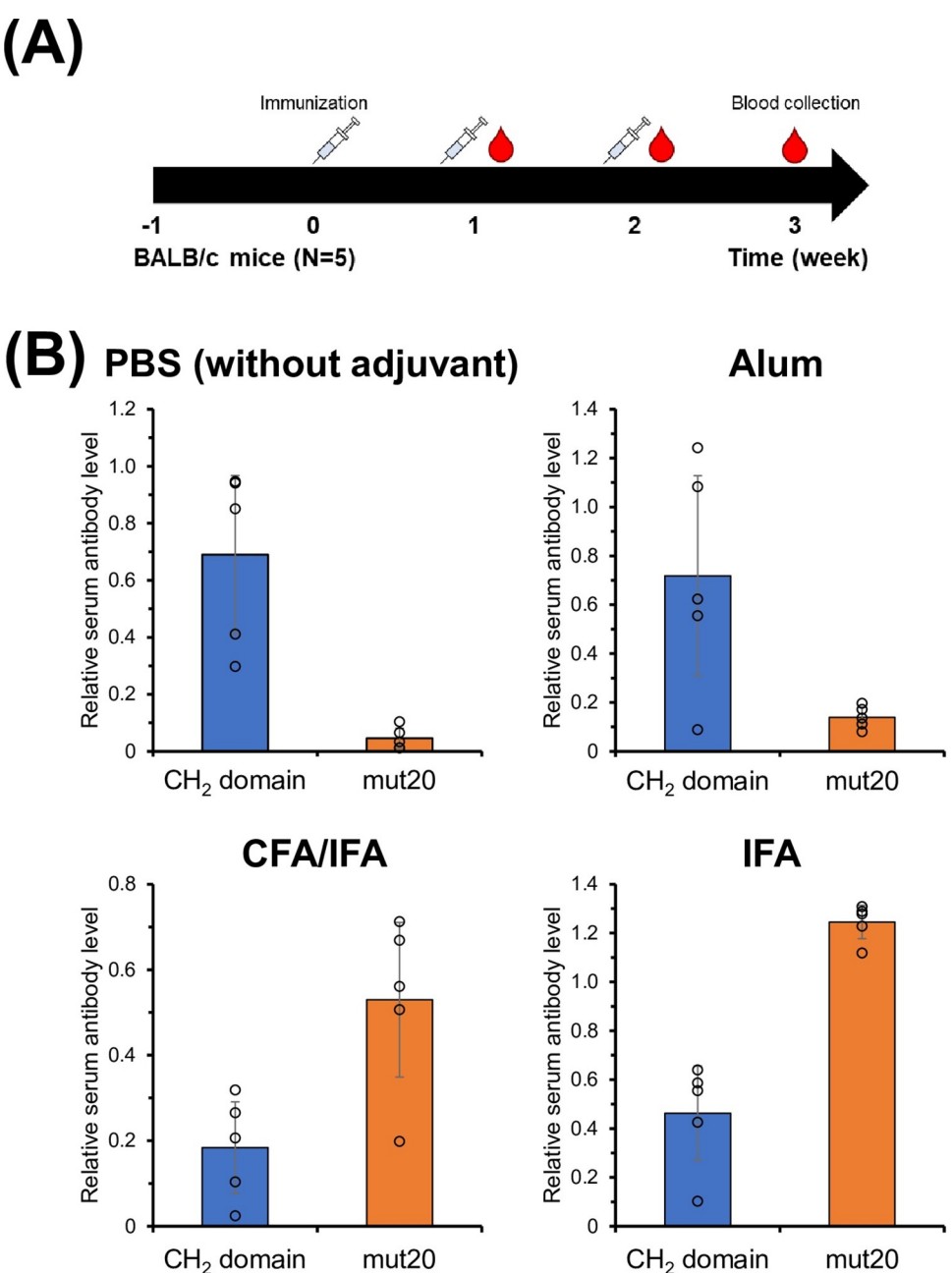

**Fig 1.** (A) Schematic representation of the immunization schedule for BALB/c mice. $CH_2$ domain and mut20 were administered into mice with either CFA/IFA or Alum adjuvant (30 μg/shot) or PBS (100 μg/shot). Mice were boosted at 1 and 2 weeks, respectively. Serum samples were collected at 1, 2, and 3 weeks after the initial vaccination. (B) ELISA analysis was performed to detect the increase of IgGs bound to the $CH_2$ domain or mut20 in the sera of vaccinated mice 3 weeks after the first immunization (N = 5). The $CH_2$ domain or mut20 was coated onto ELISA plates. The graphs represent the mean with standard deviation of individual mice from independent experiments.

9.6), at a concentration of 1 μg/ml and a temperature of 4˚C overnight. After washing with PBST, blocking was achieved with a solution containing 5% skim milk in PBST at 37˚C for one hour. After washing, 100 μl of mouse sera collected at three weeks were incubated in the plates at 37˚C for one hour. The sera were diluted to 1:50 (PBS), 1:1000 (Alum) and 1:2000 (CFA/IFA, IFA). The plates were washed and then incubated with 1:10000-diluted horseradish

peroxidase-conjugated goat Fab anti-mouse IgG solution at 37°C for one hour. After washing, 50 μl of 10.9 μM ABTS solution (2,2'-azino-bis (3-ethylbenzothiazoline-6-sulfonic acid) diammonium salt, which is a substrate for the horseradish peroxidase in 20 ml of 0.1 M citrate buffer containing 0.04 μl of 30% $H_2O_2$, pH 4.0) was added to the plates. In contrast to other substrates such as TMB (3,3',5,5'-tetramethylbenzidine), ABTS demonstrates a reduced reaction rate and is frequently employed without requiring a stop solution. The plates were incubated at 37°C and the absorbance was measured at 405 nm. To ensure measurement accuracy, the background absorbance was subtracted from the raw data prior to analysis.

## Protease digestion assay

Fifty micrograms of protein substrates were prepared to 0.5 mg/ml in 100 mM MES buffer (pH 5.0) and 5 mM mercaptoethanol. The substrates were added to 0.2 mU of cathepsin B, which is a major lysosomal cysteine protease with both endopeptidase and exopeptidase activities [39]. Digestion reactions were performed at 37°C and stopped at various time intervals by freezing the samples. After centrifugation (13,000 rpm, 4°C, 10 min), the analysis of the digested peptide in the supernatant was analyzed by size exclusion chromatography (SEC) using a column of PolyHYDROXYETHYL A column (200 Å, 5 μm, 4.6 mm × 200 mm, PolyLc inc., USA) at room temperature and a flow rate of 0.1 ml/min. The mobile phase buffer was composed of 200 mM $Na_2SO_4$ and 5 mM $KH_2PO_4$ supplemented with 25% acetonitrile (pH 3.0).

## MALDI-TOF-Mass

Protease digests were performed according to the section of "Protease digestion assay". The digestion reactions were performed at 37°C for 24 hr. After cathepsin digestion, the samples were desalted using a Zip tip C18 (Merck Millipore, USA). MALDI-TOF-Mass analysis was performed using an Autoflex III mass spectrometer (Bruker Daltonics, Germany) with sinapinic acid as a matrix reagent.

## Results

### The effect of adjuvants on the immunogenicity of the $CH_2$ domain with different conformational stabilities

To investigate if adjuvants are involved in the relationship between the stability and immunogenicity of the $CH_2$ domain, Alum and Freund's adjuvant were used. The $CH_2$ domains were expressed in *Pichia pastoris* using the methodology described in previous studies [35, 36]. The $CH_2$ domains expressed exist in two forms: glycosylated and non-glycosylated. The identification of $CH_2$ domain was determined using MALDI-TOF-MS measurements. In this study, we utilized the non-glycosylated variant, which was isolated via cation exchange chromatography, to minimize any potential impact of protein glycosylation status on the antibody induction capacity (S1 Fig). The immune response (IgG production) elicited by mut20 in mice was lower than that of the intact $CH_2$ domain when Alum was used as an adjuvant or when no adjuvant (PBS) was used (Fig 1B). On the other hand, when CFA/IFA was used as an adjuvant, the production of IgG in mice induced by mut20 was higher than that induced by the intact $CH_2$ domain. In order to alleviate the effects of inactivated *Mycobacterium tuberculosis*, which is recognized for its potent immune-stimulating properties, we conducted an animal experiment utilizing only IFA as an adjuvant. It was observed that the immune response induced by mut20 was superior to that elicited by the intact $CH_2$ domain, like the animal experiment utilizing CFA/IFA. The findings suggest that mut20 does not induce an immune response when

co-administered with Alum or PBS; however, it does elicit a response when used in conjunction with Freund's adjuvant.

### Protease digestion assay of CH$_2$ domains

Immune responses have been found to be closely linked to the resistance of proteases in APCs [12–15] because the immune response is influenced by the amount of the peptide-MHC complexes on helper T cells. To evaluate the antigen processing by APCs *in vitro*, the antigens were digested by cathepsin B, which is a prominent lysosomal protease. MALDI-TOF-Mass analysis revealed comparable proteolytic fingerprints between the intact CH$_2$ domain and mut20 following cathepsin B digestion (Fig 2). To analyze the digestion time course, the CH$_2$ domains were subjected to SEC after cathepsin digestion at various time points using a PolyHYDROX-YETHYL A column. The column successfully separated tryptic peptides of reduced and S-

**(A)**

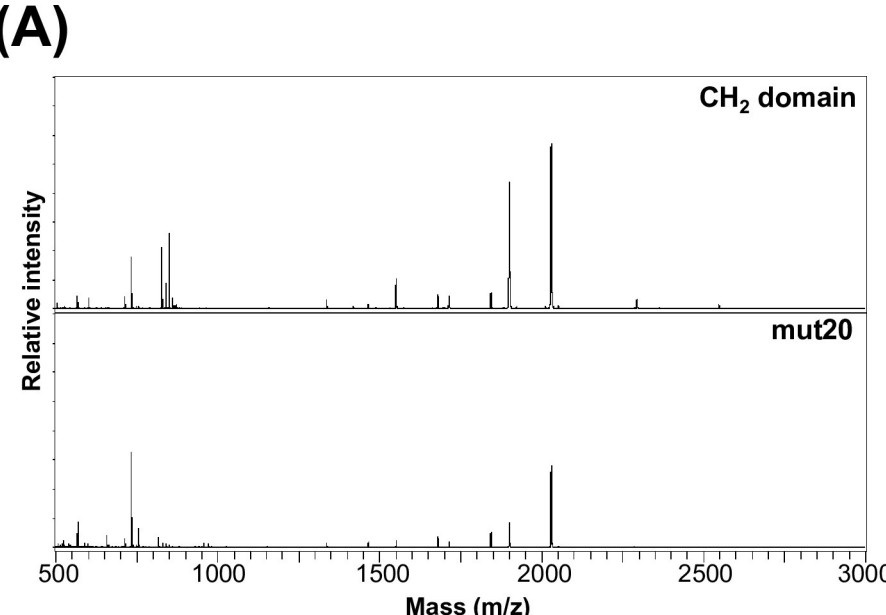

**(B)**

| CH$_2$ domain | | mut20 | |
|---|---|---|---|
| 713.595 | 1549.999 | 713.658 | 1550.132 |
| 732.573 | 1679.034 | 732.639 | 1679.159 |
| 826.634 | 1713.114 | 816.641 | 1713.217 |
| 838.656 | 1842.076 | 838.654 | 1842.228 |
| 1335.900 | 1899.135 | 1335.954 | 1899.243 |
| 1417.825 | 2028.226 | | 2028.268 |
| 1464.910 | 2546.932 | 1465.030 | 2546.932 |

**Fig 2.** (A) MS chromatograms of the CH$_2$ domain and mut20. The digestion reaction with cathepsin B was conducted at 37°C for 24 hours. The peptides were analyzed using MALDI-TOF-Mass. (B) The mass of peptides is depicted in Fig 2A. The masses of the CH$_2$ domain and mut20 were essentially identical. The discrepancy of 9.993 between 826.634 and 816.641 could be attributed to the substitution of leucine and cysteine as predicted by the peptide mass analysis.

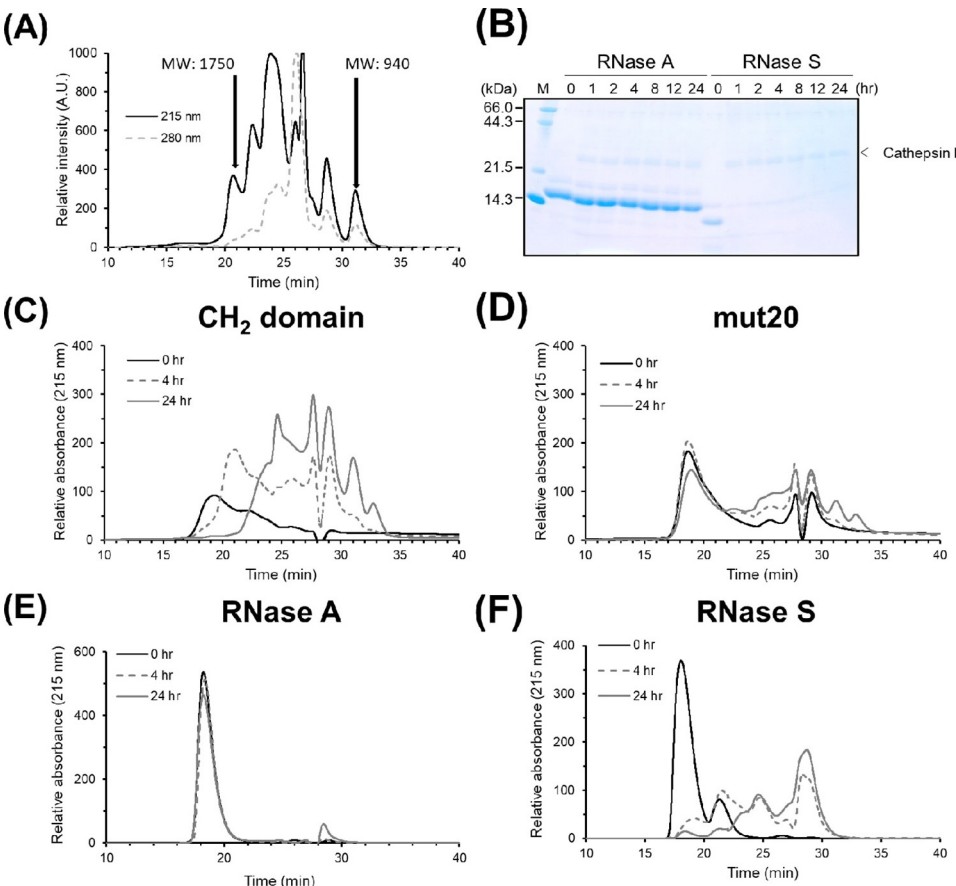

**Fig 3. Protein digestion assay by cathepsin B.** (A) SEC patterns of the tryptic peptides of reduced and S-carboxymethylated hen lysozyme were monitored using a UV detector at 215 nm and 280 nm. The elution times of the tryptic peptides with molecular weights of 1750 and 940 are indicated. (B) SDS-PAGE results of RNase A and RNase S after cathepsin B digestion. The levels of RNase A and RNase S were analyzed after incubation with cathepsin B at pH 5.0 and 37˚C for a suitable duration. SDS-PAGE followed by coomassie brilliant blue staining was used for analysis. (C)–(F): SEC patterns of the digested peptides from the protein were monitored at pH 5.0 and 37˚C using cathepsin B for 0 hr (black solid line), 4 hr (gray broken line) and 24 hr (gray solid line). The monitoring was done using a UV at 215 nm. (C) was CH2 domain, (D) was mut20, (E) was RNase A and (F) was RNase S, respectively.

carboxymethylated hen lysozyme. The molecular weights of 940 and 1750 corresponded to peptides of seven and sixteen amino acids in lengths, respectively (Fig 3A). Since the crystallographic analysis has revealed that MHC class II on T cells can bind at least octamer peptides when presenting to T cell receptors [40], the column could potentially be used to analyze the quantities of peptides that can bind to the MHC class II receptor on T cells. The intact CH2 domain was digested by cathepsin B at a faster rate compared to mut20 (Fig 3C and 3D), suggesting that mut20 is more resistant to the lysosomal protease activity of cathepsin B. Therefore, the differences in immunogenicity of CH2 domains cannot be solely explained by the amounts of MHC-peptide complexes.

## Protease digestion assay of ribonucleases

The study employed RNase A and RNase S to examine the correlation between immunogenicity and peptide levels in proteins. The results differed from those observed with CH2 domains when Alum was utilized as the adjuvant. Namely, previous studies have shown that RNase A is

more immunogenic than RNase S (destabilized form) [19, 20]. As shown in Fig 3B, 3E and 3F, RNase A exhibited greater resistance to protease digestion compared to RNase S due to its higher conformational stability. The amount of digestion peptides from RNase S was higher than that from RNase A, which was consistent with the results in the previous report [19, 20]. There is a prevailing belief that an elevated abundance of MHC-peptide complexes results in heightened antibody production. Given that the sequences of RNase A and S are nearly identical, with the only difference being the presence of a nick, it was initially expected that RNase A would produce a greater number of peptides. Contrary to expectations, RNase S produced a notably higher quantity of peptides. Moreover, RNase A exhibited minimal cleavage into peptides, suggesting that the immunogenicity is not solely determined by the extent of protease digestion.

## Discussion

The impact of protein stabilization on immunogenicity has been a subject of controversy [4, 13–20]. We then examined how different adjuvants affect the relationship between protein stability and its immunogenicity. To evaluate the influence of protein stability on immunogenicity while maintaining consistent adjuvant conditions, we employed varying protein dosages in groups with and without adjuvants to guarantee equivalent levels of antibody production. This methodology was employed to evaluate the distinct immune stimulation capabilities of various adjuvants and enable precise comparisons among heterogeneous groups. When employing a CFA containing inactivated mycobacteria in animal studies, despite its benefits in boosting immune responses [24, 25], it is associated with several notable limitations, especially concerning animal welfare concerns. Moreover, adding to the complexity, CFA has been employed as an adjuvant initially, followed by IFA without inactivated mycobacteria in the animal experimental procedure. While acknowledging the potential limitation of this approach in enabling direct comparisons of antibody responses among different adjuvant groups due to variations in protein dosage, it does enable the evaluation of the relative immunogenicity of proteins with varying stability profiles under standardized adjuvant conditions. The increase in antigen dosage is commonly recognized to result in an enhanced production of antibodies, a conclusion supported by our previous study [21].

Furthermore, we are focusing on peptides that can bind to the MHC class II receptor. Therefore, employing multiple mouse strains with diverse MHC types would offer greater advantages. Tsujihata et al. identified a crucial single-point mutation located within the T cell epitope of hen egg lysozyme, which plays a vital role in T cell recognition [41]. This mutation resulted in a limited production of IgG antibodies that specifically recognize the antigen. Based on the report, if the two-point mutations between the $CH_2$ domain and mut20 are located within the T cell epitope of the $CH_2$ domain, it is hypothesized that there will be a minimal IgG response subsequent to the administration of antigens to mice using Alum or CFA/IFA. It was considered likely that the mutation sites in mut20 were not situated within the T cell epitope of the $CH_2$ domain, as indicated by the consistent production of IgG observed across all experimental conditions involving animals. In this investigation, animal experiments were specifically carried out on BALB/c mice, with careful attention given to animal welfare considerations.

Immunization of intact $CH_2$ domain mixed with Alum or just PBS resulted in a stronger IgG response compared to the stabilized form (Fig 1). The amount of digestion peptides from $CH_2$ domain was higher than that from mut20 (Fig 3C and 3D). We have confirmed that stabilizing the $CH_2$ domain reduces immunogenicity by inhibiting protease digestion using cathepsin B. This finding may help reduce the immunogenicity of therapeutic antibodies by

addressing the instability of the $CH_2$ domain compared to other domains in human antibodies.

When employing CFA/IFA adjuvant and IFA adjuvant alone, mice with the intact $CH_2$ domain exhibited a diminished immune response in comparison to mut20 (Fig 1). Hence, the comparable outcomes obtained for both CFA/IFA and IFA indicate that the variances noted were not influenced by inactivated *Mycobacterium tuberculosis* but rather by the effect of the adjuvant on protein stability.

In this case, the immune response cannot be solely explained by the amount of peptide digested by cathepsin B. It has been reported that Freund's adjuvant can decrease the conformational stability of antigens because of its hydrophobic nature [26, 27]. Therefore, in order to elicit an immune response in moderately stable proteins such as the $CH_2$ domain (with a Tm of 54.1˚C at pH 5.5 [42]), protein stabilization may be necessary to counteract destabilization caused by interactions with Freund's adjuvant. On the other hand, in our prior study [21], the administration of mice with the SARS-CoV-2 spike protein using CFA/IFA adjuvant did not result in the production of neutralizing antibodies. The trimeric spike protein undergoes hinge-like conformational movements that transiently hide or expose the receptor binding domains (RBDs) and the predominant state of the trimer, which facilitates evasion to cells, involves one of the three RBDs being rotated upwards in the receptor-accessible conformation, as demonstrated using Cryo-EM [43]. Furthermore, Cryo-EM analysis revealed that a neutralizing antibody was bound to one of the three RBD and caused it to rotate upwards [44]. Hence, it is plausible that the destabilizing impact caused by mineral oil from CFA or IFA could influence the hinge-like conformational changes of the RBD in the trimeric S-protein.

Previous studies have shown that the immune responses in mice were reduced when protein antigens with higher Tm (such as hen lysozyme, Phlp 7, Der p2 and toxin α) had increased conformational stabilities, even when using CFA/IFA or Alum as adjuvants [14–19] (as shown in the case of "high stability" in Fig 4). According to reports, hen lysozyme has a Tm of 75˚C (at pH 4.0, [45]), Phlp 7 has a Tm of 77.3˚C (at pH 7.4, [46]), Der p2 has a Tm above 70˚C (at pH 5.0, [47]) and toxin α has a Tm above 85˚C (at pH 7.0, [14]) (Table 1). On the other hand, RNase S has a Tm of 46.8˚C (at pH 6.0, [48]) and apo-HRP has a Tm below 37.3˚C (at pH 7.0, [49]), indicating that these proteins were intrinsically less stable than the $CH_2$ domain employed in this study (Table 1). It was reported that the immune responses in mice increased when protein antigens with lower melting temperatures (such as RNase S and apo-HRP) were

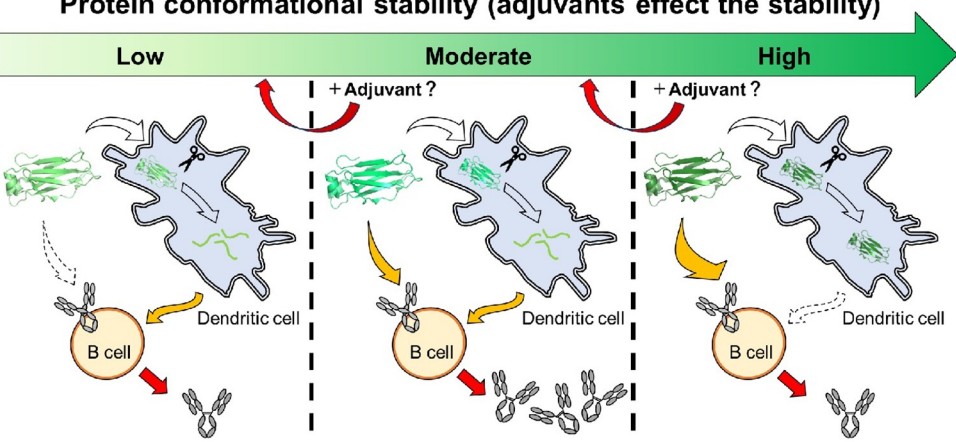

**Fig 4. New interpretations on the relationship between protein conformational stability and antibody production.**

**Table 1. Relationship between protein stability and its immunogenicity when administering antigens to mice using adjuvants.**

| Antigen | | | Conformational stability | | Immunogenicity | |
|---|---|---|---|---|---|---|
| Origin | Perturbation of stability | Description | Tm (˚C)*1 | $C_{1/2}$ (M)*2 | Adjuvant | Antibody production |
| Bovine ribonuclease | | RNase A | 62.8 (7.0)[48] | | Alum | RNase S |
| | limited proteolysis | RNase S | 46.8 (6.0)[48] | | | < RNase A[19] |
| Horseradish peroxidase | | Holo-HRP | 77.8 (7.0)[49] | | Alum | Apo-HRP |
| | removal of heme | Apo-HRP | 37.2 (7.0)[49] | | | < Holo-HRP[19] |
| Hen egg white lysozyme | | Lysozyme | 75.0 (4.0)[45] | | CFA/IFA | Lysozyme < |
| | cross-linking | 35–108 CL lysozyme | 85.4 (3.0)[17] | | | 35–108 CL lysozyme[17] |
| Grass pollen allergen Phl p 7 | | Phlp 7 | 77.3 (7.4)[46] | 2.2[17] | CFA/IFA | M5C Phlp7 |
| | cross-linking | M5C Phlp 7 | ND*3 | 5.3[17] | | < Phlp7[17] |
| Major house dust mite allergen | | Der p2 | 70 (5.0)[47] | 1.9[18] | Alum | A122I Der p2 |
| | cavity-filling | A122I Der p2 | ND*3 | 2.8[18] | | < Der p2[18] |
| CH2 domain in human antibody | | CH2 domain | 54.1 (5.5)[42] | | None | mut 20 |
| | cross-linking | mut 20 | 72.8 (5.5)[42] | | | < CH2 domain*4 |
| | | CH2 domain | 54.1 (5.5)[42] | | Alum | mut 20 |
| | cross-linking | mut 20 | 72.8 (5.5)[42] | | | < CH2 domain*4 |
| | | CH2 domain | 54.1 (5.5)[42] | | CFA/IFA | CH2 domain |
| | cross-linking | mut 20 | 72.8 (5.5)[42] | | | < mut 20*4 |

*1Thermal stability.

*2 $C_{1/2}$ refers to the guanidine concentration at which half of the protein is denatured under conditions that assess the conformational stability of a protein for the unfolding by guanidine hydrochloride.

*3"ND" indicates the absence of any reported incidents to date.

*4The data presented in this report.

administered with higher conformational stabilities, even when Alum was used as an adjuvant [19, 20].

When a B cell receptor (BCR) encounters an antigen with a compatible conformation, the antigen binds to a specific site on the BCR, similar to a lock and key mechanism. Based on our findings, we have demonstrated that a protein antigen must possess a minimum level of intrinsic conformational stability to bind with BCR effectively and induce an immune response (as shown in the case of "low stability" in Fig 4). Indeed, the tertiary structure of RNase S begins to unfold at 37˚C at pH 6.8, as observed by near-UV CD [50]. Therefore, when the population of tertiary structure in RNase S was lower in mice, it was not always able to effectively bind the corresponding BCR, resulting in reduced immunogenicity. In order to enhance the immune response, it may be beneficial to improve the conformational stability of a protein antigen that is unstable in mice body when administering antigens to mice.

In summary, it is crucial to note that the optimal stability of antigens for antibody production is expected to vary depending on the utilization of various adjuvants in animal research (Table 1). To enhance antibody production, it is recommended to utilize customized strategies that are consistent with the stability profile of each specific protein. In cases where proteins demonstrate significant instability, such as RNase S and apo-HRP, it is recommended to adopt an approach to achieve stabilization and maintain structural integrity, as depicted in the example of "low stability" in Fig 4). Conversely, for proteins with high stability, such as lysozyme, deliberate destabilization may be beneficial in augmenting immune responses, as depicted by the "high stability" scenario depicted in Fig 4. In the case of proteins with moderate stability, such as the CH2 domain, the choice between stabilization and destabilization should be carefully tailored to the particular adjuvant employed, as demonstrated in the "moderate stability"

case in [Fig 4](). This tailored approach recognizes the unique interplay between protein stability and adjuvant properties, aiming to optimize the conditions for antibody production.

## Supporting information

**S1 Fig. Purification of glycosylated CH2 domain.** The $CH_2$ domain was expressed in *Pichia pastoris* and purified following the protocol described by Oyama et al. (BBRC, 2021). In summary, the supernatant underwent purification using a TOYOPEARL SP-650M cation exchange chromatography column (2.6 cm × 2 cm). Moreover, the protein fractions were further purified using a TOYOPEARL SP-650M cation exchange chromatography column (1.6 cm × 100 cm). The eluted fractions were monitored at 280 nm (A) and analyzed utilizing SDS-PAGE (B). The study carried out by Oyama et al. (J. Biochem., 2021) demonstrated that the molecular weight of the expressed protein corresponded to the theoretical weight of the $CH_2$ domain. This discovery confirms that the expressed protein accurately represents the $CH_2$ domain. While these figures show similarities to those illustrated in the research conducted by Oyama et al. (J. Biochem. 2021), these figures differed from those reported in the study conducted by Oyama et al. (J.Biochem., 2021).
(JPG)

**S1 Data. Animal experiment raw data.**
(XLSX)

## Acknowledgments

We would like to thank Dr. Takahashi and Prof. Caaveiro from Kyushu University for their kind guidance in the experimental procedure and Prof. Maenaka from Hokkaido University for his valuable comments during the preparation of the revised manuscript.

## Author Contributions

**Conceptualization:** Kosuke Oyama, Tadashi Ueda.

**Data curation:** Kosuke Oyama.

**Project administration:** Tadashi Ueda.

**Writing – original draft:** Kosuke Oyama.

**Writing – review & editing:** Kosuke Oyama, Tadashi Ueda.

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
