## [Decision Letter · Decision Letter 0]

5 Mar 2024

PONE-D-23-43523Relationship between protein conformational stability and its immunogenicity when administering antigens to mice using adjuvants - Analysis employed the CH2 domain in human antibodies -PLOS ONE

Dear Dr. Ueda,

Thank you for submitting your manuscript to PLOS ONE. After careful consideration, we feel that it has merit but does not fully meet PLOS ONE’s publication criteria as it currently stands. Therefore, we invite you to submit a revised version of the manuscript that addresses the points raised during the review process.

We look forward to receiving your revised manuscript.

Kind regards,

Rui Tada, Ph.D.

Academic Editor

PLOS ONE

Journal Requirements:

2. To comply with PLOS ONE submissions requirements, in your Methods section, please provide additional information regarding the experiments involving animals and ensure you have included details on (a) methods of sacrifice, (b) methods of anesthesia and/or analgesia, and (c) efforts to alleviate suffering.

This work was supported by the MEXT (Ministry of Education, Culture, Sports, Science, and Technology) Grant, Kyushu University operating expenses, and under the “COVID-19 Drug and Vaccine Development Donation Account” Project from Sumitomo Mitsui Trust Bank.

This work was supported by the MEXT (Ministry of Education, Culture, Sports, Science, and Technology) Grant, Kyushu University operating expenses, and under the “COVID-19 Drug and Vaccine Development Donation Account” Project from Sumitomo Mitsui Trust Bank.  We would like to thank Dr. Takahashi and Prof. Caaveiro from Kyushu University for their kind guidance in the experimental procedure.

This work was supported by the MEXT (Ministry of Education, Culture, Sports, Science, and Technology) Grant, Kyushu University operating expenses, and under the “COVID-19 Drug and Vaccine Development Donation Account” Project from Sumitomo Mitsui Trust Bank.

5. We note that your Data Availability Statement is currently as follows: All relevant data are within the manuscript.

Reviewers' comments:

Reviewer's Responses to Questions

**Comments to the Author**

1. Is the manuscript technically sound, and do the data support the conclusions?

Reviewer #1: Yes

Reviewer #2: Partly

2. Has the statistical analysis been performed appropriately and rigorously? 

Reviewer #1: Yes

Reviewer #2: No

3. Have the authors made all data underlying the findings in their manuscript fully available?

Reviewer #1: Yes

Reviewer #2: Yes

4. Is the manuscript presented in an intelligible fashion and written in standard English?

Reviewer #1: Yes

Reviewer #2: Yes

5. Review Comments to the Author

Reviewer #1: The work "Relationship between protein conformational stability and its immunogenicity when administering antigens to mice using adjuvants - Analysis employed the CH2 domain in human antibodies"carefully describes the importance of the stability of antigens for antibody production that vary depending on the use of different adjuvants .Another point to remember is that different proteins require stabilisation to activate the immune response. The role of the djuvant used is effective in stimulating mediated immunity both for the potentiation of T helper cells that lead to the production of different types of immunoglobulins and the effective participation of effector T cells.The authors also point out that it is important to note that the stability of the antigens used in an immunisation scheme will have to be standardised, the antigens for the production of antibodies and in the case of mice the importance of the strain chosen for the type of response expected. In previous studies "Zhang N, Channappanavar R, Ma C, Wang L, Tang J, Garron T, et al. evaluated adjuvants for receptor-binding domain-based subunit vaccines against Middle East respiratory syndrome virus". Cell Mol Immunol. 2016;13: 180-190. doi:10.1038/cmi.2015.03 thus showing the importance of the adjuvant in modulating the immune response.

The experimental design is adequate, but there is some error in the text.

Please review

line 118 PolyHYDROXYETHYL A column (200 Å,

line 126 using an AutoflexÅ mass spectrometer (Bruker Daltonics, Germany) w

Reviewer #2: General comment: The authors have made an attempt to elucidate the stability of proteins and their immunogenicity. Overall, the rationale and study design are quite interesting. However, further clarification and improvements in manuscript writing are needed.

Specific comments:

Abstract:

1. The abstract should be based solely on your findings, e.g., anti-drug antibodies are not directly related to this study.

2. The abstract should not include discussion points, such as “The findings, in conjunction with a previous study, indicate that protein stability plays a crucial role in eliciting an immune response in mice through the binding of protein antigens to B cell receptors on APCs.”

3. There is no supporting data from this study for “It is important to note that the optimal stability of antigens for antibody production is anticipated to vary depending on the use of different adjuvants in animal experiments.”

Introduction:

4. The mechanism of action of Alum and IFA/CFA is required, as they might have different immune stimulation pathways, which could lead to differences in immune induction capability (not solely due to the effect of protein stability).

5. RNase A and RNase S were not mentioned.

6. Information about mut20 (mutation position, changes in disulfide bonds, etc.) and its conformation (compared to the wild-type) must be included.

Materials and Methods:

7. Producing mut20 in yeast could potentially yield a glycosylated protein. How did the authors avoid the effect of the glycosylation status of these two proteins on antibody induction capability?

8. The statement “All other chemicals used in the study were of the highest commercial quality available” is vague. How can the authors ensure they were the best quality on the market?

9. The immunization schedule was quite short (1-week intervals). For protein (and other platform) vaccine immunization, at least a 2-week interval is usually employed to allow immune cell maturation.

10. The approved animal protocol number should be included.

11. Why was the intraperitoneal route used instead of the intramuscular route? This does not reflect the standard immunization procedure.

12. Why were different protein amounts used for adjuvanted and non-adjuvanted groups (30 vs. 100 micrograms)? We cannot directly compare the magnitude of the antibody response as the vaccine doses differ.

13. What is the reason for using CFA in the first immunization and IFA for the booster? Please explain and state this in the manuscript.

14. Salt concentrations in buffers must be indicated as their final concentrations (mol/liter, molar), not in grams.

15. Line 105, what is the initial dilution, and how did you dilute the tested serum?

16. ELISA: The methods for stopping the reaction and reporting readouts are missing. Was the background subtracted?

Results:

17. For all results sections, please do not repeat the introduction and method. Authors must directly explain the obtained results concisely (important). Also, the discussion should be stated separately in the discussion section.

18. For better understanding, CH2, mut20, RNase A, and RNase S should be indicated in Figures 3C-3F.

19. Again, authors never mentioned and explained about RNase A/RNase S; it suddenly appeared in lines 165-177.

20. Where is the data to support the statement? “When examining the MHC-peptide complex in APCs and T cell presentation, it is suggested that the immunogenicity of RNase A compared to RNase S is not solely determined by the degree of protease digestion of the RNases.”

Discussion:

21. More discussion is needed on topics other than protein stability, including adjuvant properties. Moreover, as the authors tested in BALB/c mice, which is an inbred strain with a restricted TCR and MHC repertoire, further investigation in an outbred mouse strain may be needed to confirm your findings.

22. What is the explanation for the opposite IgG results when the adjuvant is changed (Figure 1B)?

6. PLOS authors have the option to publish the peer review history of their article (what does this mean?). If published, this will include your full peer review and any attached files.

Reviewer #1: No

Reviewer #2: **Yes: **Eakachai Prompetchara

---

## [Author Response · Author response to Decision Letter 0]

10 Apr 2024

Specific comments: 

(Abstract)

Comment 1

The abstract should be based solely on your findings, e.g., anti-drug antibodies are not directly related to this study. 

Response: Based on your comment, we have revised the abstract to focus solely on the findings directly related to this study. We have removed the sentences about the anti-drug antibodies.

Comment 2

The abstract should not include discussion points, such as “The findings, in conjunction with a previous study, indicate that protein stability plays a crucial role in eliciting an immune response in mice through the binding of protein antigens to B cell receptors on APCs.” 

Comment 3

There is no supporting data from this study for “It is important to note that the optimal stability of antigens for antibody production is anticipated to vary depending on the use of different adjuvants in animal experiments.”

Response: Thank you for your comments. We have revised the abstract accordingly to remove discussion points. Instead, the findings of this study have been incorporated into the Abstract section of the revised manuscript.

(Introduction)

Comment 4

The mechanism of action of Alum and IFA/CFA is required, as they might have different immune stimulation pathways, which could lead to differences in immune induction capability (not solely due to the effect of protein stability). 

Response: Thank you for your valuable feedback. We comprehended the supplementary explanation regarding the mechanism of action of Alum and IFA/CFA, because the outcome of this study was found to be associated not only with the impact on protein stability but also with a biological factor, specifically the mechanism of action, of CFA/IFA. In the revised manuscript, we have incorporated a description on the mechanism of action of Alum and CFA/IFA, along with an analysis of the impact of CFA/IFA on protein stability, into the text (page 3 - page 4 line 62 - line 76 in the Introduction section). Based on the literature (Biliau and Matthys, J. Leukoc. Biol., 2001; Lim, Int. Immunopharmacol., 2003) referenced in the revised manuscript, it has been documented that the presence of inactivated Mycobacterium tuberculosis in CFA influences immunological responses. This includes the stimulation of T-lymphocytes to adopt a Th1 profile, leading to a robust delayed type hypersensitivity reaction against autoantigens and an upregulation of Toll-like receptor 2 expression. Hence, to eliminate the impact of inactivated Mycobacterium tuberculosis on IgG expression following the administration of an antigen with an adjuvant, we conducted a supplementary series of animal experiments utilizing only IFA as the adjuvant. Consequently, our findings indicate that the production of IgG against mut20 was higher compared to that of CH2 domain when administered with IFA as an adjuvant, suggesting that a pattern that was consistent when administering them with CFA/IFA. Therefore, our results suggest that the variations in IgG production observed during antigen administration with Alum versus CFA/IFA were not influenced by the biological activity of inactivated Mycobacterium tuberculosis. Instead, they were attributed to the impact of the adjuvant on protein stability. Therefore, the revised manuscript includes the outcomes of a sequence of animal experiments that employed IFA solely as an adjuvant (see Figure 1B and page 9, line 163 - line 169) in the revised manuscript. Additionally, the aforementioned description, in conjunction with the references, has been incorporated into the Discussion section of the revised manuscript (page 11, lines 212-215) .

Comment 5

RNase A and RNase S were not mentioned. 

Response: We acknowledge the feedback provided regarding the description of RNase A and RNase S used in our study. Delamarre et al. (Science, 2005; J. Exp. Biol. Med. 2006) showed that the administration of RNase A with Alum in mice led to increased IgG production in comparison to the administration of RNase S. Additionally, it was observed that RNase A exhibited greater stability than RNase S. These results seem to conflict with the outcomes of our investigation on the CH2 domain in animal experiments involving Alum. In response to your inquiry, information concerning RNase A and RNase S has been integrated into the introduction (page 5, line 89 - line 100). By incorporating these proteins in our research, we can explore the importance of conformational stability on the immune response when administering protein antigens to mice with Alum adjuvant.

Comment 6

Information about mut20 (mutation position, changes in disulfide bonds, etc.) and its conformation (compared to the wild-type) must be included.

Response: Thank you for acknowledging the importance of providing thorough information on mut20 and its conformational changes in relation to the wild-type protein. In our response, we have furnished comprehensive details concerning mut20, including its mutation position and alterations in disulfide bonds, in the pertinent section of the manuscript (page 4, line 81 - line 83) in the revised manuscript.

(Materials and Methods)

Comment 7

Producing mut20 in yeast could potentially yield a glycosylated protein. How did the authors avoid the effect of the glycosylation status of these two proteins on antibody induction capability? 

Response: We appreciate the constructive feedback received on our manuscript. In numerous instances, glycosylated and non-glycosylated proteins are acquired through the yeast expression system. In our prior investigation (Oyama et al., J Biochem, 2021; Biophys. Biochem. Res. Commun, 2021), we effectively isolated the glycosylated and non-glycosylated CH2 domain through cation exchange chromatography. In this study, we utilized the non-glycosylated protein to mitigate any potential impact of protein glycosylation status on antibody induction capacity.

Cation exchange chromatography of CH2 domain. 

Comment 8

The statement “All other chemicals used in the study were of the highest commercial quality available” is vague. How can the authors ensure they were the best quality on the market? 

Response: Based on your feedback, we altered the sentence of “All other chemicals used in the study were of the highest commercial quality available” to that of “All the other reagents and chemicals used in the study were of analytical grade or higher quality” (page 6, line 107 - line 108) in the revised manuscript.

Comment 9

The immunization schedule was quite short (1-week intervals). For protein (and other platform) vaccine immunization, at least a 2-week interval is usually employed to allow immune cell maturation.

Response: Thank you for your valuable feedback. We recognize that our immunization schedule, with 1-week intervals, may appear shorter when compared to the conventional practice, which typically involves a 2-week interval for protein vaccine immunization. Our immunization protocol was developed based on methodologies previously established in our laboratory (Ohkuri et al., J. Immunol, 2010; Nakamura et al., Biochim. Biophys. Acta, 2016, Ohkuri et al., Biochem. Biophys. Rep., 2019, and Masuda et al., Front. Immunol., 2022). Despite the unconventional schedule, this protocol has consistently produced monoclonal IgGs with strong binding abilities, resulting in a sub-nanomolar KD of the Fab-antigen complex during the hybridoma generation process. Despite concerns about immune cell maturation, the generation of high-affinity antibodies suggests that immune cells have undergone effective maturation in accordance with our immunization protocol. This observation indicates that the effectiveness of our methodology, specifically the relatively short interval of immunization (1-week intervals), has little effect on inducing strong immune responses.

Comment 10

The approved animal protocol number should be included. 

Response: Based on your comment, we have included the number, A21-053-0 (line 115 on page 6) in the revised manuscript.

Comment 11

Why was the intraperitoneal route used instead of the intramuscular route? This does not reflect the standard immunization procedure. 

Response: Thank you for your valuable feedback. The intraperitoneal route was selected based on several considerations. Firstly, it has been widely utilized in previous research studies that investigate similar research questions, thereby enhancing the ability to compare and maintain consistency across experiments. Furthermore, Freund's adjuvant, which was employed in our study, is recommended for intraperitoneal administration in the manufacturer's guidelines. Moreover, Delamarre et al. (2006) illustrated in their research published in the Journal of Experimental Medicine, which paper is closely related in this study, that intraperitoneal administration resulted in increased antibody production in comparison to intramuscular administration and was also observed that the method of administration did not have a significant impact on protein stability and immunogenicity. We think that these suggests that the intraperitoneal route can be considered a suitable option without compromising the integrity of our experimental results.

Comment 12

Why were different protein amounts used for adjuvanted and non-adjuvanted groups (30 vs. 100 micrograms)? We cannot directly compare the magnitude of the antibody response as the vaccine doses differ. 

Response: We appreciate the constructive feedback received regarding our manuscript. The main aim of our study was to evaluate the influence of protein stability on immunogenicity while maintaining consistent adjuvant conditions. To accomplish this, varying protein dosages were utilized for the adjuvanted and non-adjuvanted groups to guarantee equivalent levels of antibody production. This approach was employed to consider the unique immune stimulation capacities of various adjuvants and to enable valid comparisons between different groups. While acknowledging that this approach may restrict direct comparisons of antibody responses among different adjuvant groups because of variations in protein dosage, it enables us to evaluate the relative immunogenicity of proteins with different stability levels under standardized adjuvant conditions.

Comment 13

What is the reason for using CFA in the first immunization and IFA for the booster? Please explain and state this in the manuscript. 

Response: Thank you for your comment. This sequential administration adheres to a standard protocol in which CFA, recognized for its ability to induce strong inflammation, is used initially, followed by IFA for enhancing immune responses.

Comment 14

Salt concentrations in buffers must be indicated as their final concentrations (mol/liter, molar), not in grams. 

Response: Based on the feedback provided, we converted grams to molar or percentage units, highlighting characters in questions in red color.

Comment 15

Line 105, what is the initial dilution, and how did you dilute the tested serum? 

Response: We apologize for the lack of clarity concerning the dilution factor utilized in ELISA. In our investigation, serum samples from animals underwent varying dilution factors in ELISA to accommodate potential discrepancies in antibody production across distinct adjuvant groups. The methodology section has been revised as indicated (page 7 line 130 - line 131) in the revised manuscript.

Comment 16

ELISA: The methods for stopping the reaction and reporting readouts are missing. Was the background subtracted?

Response: In our ELISA protocol, we employed ABTS (2,2'-azino-bis(3-ethylbenzothiazoline-6-sulfonic acid)) as the substrate for the horseradish peroxidase (HRP). In contrast to other substrates like TMB (3,3',5,5'-tetramethylbenzidine), ABTS is characterized by a slower reaction rate and is commonly employed without the requirement of a stop solution. Consequently, the reaction was permitted to continue until the intended endpoint was achieved. To guarantee precision in our measurements, the background absorbance was subtracted from the raw data before conducting the analysis.

(Results) 

Comment 17

For all results sections, please do not repeat the introduction and method. Authors must directly explain the obtained results concisely (important). Also, the discussion should be stated separately in the discussion section. 

Response: We are grateful for the constructive feedback provided on our manuscript. Based on your feedback, we have removed redundant sections and focused solely on presenting the obtained results concisely.

Comment 18

For better understanding, CH2, mut20, RNase A, and RNase S should be indicated in Figures 3C-3F. 

Response : We acknowledge the recommendation to provide labels for CH2 domain, mut20, RNase A and RNase S in Figures 3C-3F to enhance clarity. 　The figures have been updated accordingly.

Comment 19

Again, authors never mentioned and explained about RNase A/RNase S; it suddenly appeared in lines 165-177. 

Response: Thank you for the valuable feedback that was provided.　As described in response to comment 5, we employed RNase A and RNase S.　 These proteins are known to produce results that differ from those of the CH2 domain in animal studies involving Alum. The aim of this study is to investigate RNase A and RNase S, as mentioned in the Introduction and result section (page 10, line 190- line 191) in the revised manuscript.

.

Comment 20

Where is the data to support the statement? “When examining the MHC-peptide complex in APCs and T cell presentation, it is suggested that the immunogenicity of RNase A compared to RNase S is not solely determined by the degree of protease digestion of the RNases.”

Response: We appreciate the constructive feedback received on our manuscript. We have included the comprehensive description (page 10, line 196 - line 202) in the revised manuscript.

.

(Discussion) 

Comment 21

More discussion is needed on topics other than protein stability, including adjuvant properties. Moreover, as the authors tested in BALB/c mice, which is an inbred strain with a restricted TCR and MHC repertoire, further investigation in an outbred mouse strain may be needed to confirm your findings. 

Response: We appreciate the constructive feedback received on our manuscript. We have incorporated the mechanism of action of Alum and CFA/IFA adjuvant as described in Introduction (page 3 - page 4, line 61 - line 75) in the revised manuscript. Moreover, in this study, we included the results of animal studies utilizing only IFA as an adjuvant to further clarify the impact of inactivated Mycobacterium tuberculosis on the immune response (Fig. 1B). We have comprehended the significance of your comment suggesting that additional research in an outbred mouse strain may be necessary to validate our findings, because we use mut20, which contains two amino acid residues that differ from the CH2 domain. Regrettably, our academic department does not possess the essential resources required to conduct animal experiments utilizing outbred mouse strains. Under the specified circumstances, it was observed that the deamidation of asparagine within the T cell epitope of hen egg lysozyme significantly influenced T cell recognition (McAdam et al. J. Exp. Med. 2001). Moreover, Tsujihata et al. (Mol. Immunol. 2001) identified a crucial single point mutation in the T cell epitope of hen egg lysozyme that is necessary for T cell recognition. This mutation resulted in a restricted generation of IgG antibodies targeting the antigen. Based on these reports, when the two point mutations between CH2 and mut20 are located within the T cell epitope of the CH2 domain, it is hypothesized that there will be a minimal IgG response subsequent to the administration of antigens to mice utilizing Alum or CFA/IFA. However, the production of IgG was observed to investigate the relationship between protein conformational stability and immunogenicity without the need to use an outbred mouse strain.

In the original manuscript, our study focused on examining the influence of the intrinsic global structure that can efficiently bind to B-cell receptors (BCR) on APCs. In a prior study conducted by Ma

---

## [Decision Letter · Decision Letter 1]

22 May 2024

PONE-D-23-43523R1Relationship between protein conformational stability and its immunogenicity when administering antigens to mice using adjuvants - Analysis employed the CH2 domain in human antibodies -PLOS ONE

Dear Dr. Ueda,

Thank you for submitting your manuscript to PLOS ONE. After careful consideration, we feel that it has merit but does not fully meet PLOS ONE’s publication criteria as it currently stands. Therefore, we invite you to submit a revised version of the manuscript that addresses the points raised during the review process.

Thank you for submitting your revised manuscript. We appreciate the effort you have put into addressing the comments from Reviewer 2.

Unfortunately, despite multiple invitations and reminders, we did not receive a response from Reviewer 1 during this round of revisions. However, upon careful examination of Reviewer 1's initial comments, we found them to be minor in nature and have determined that you have already adequately addressed them in your revised manuscript.

Considering the thoroughness of your revisions in response to Reviewer 2's comments and the minor nature of Reviewer 1's suggestions, we believe that your manuscript is now suitable for publication after addressing the remaining minor points raised by Reviewer 2.

Please submit your revised manuscript, along with a detailed response letter addressing the comments from Reviewer 2. We look forward to receiving your revised manuscript.

We look forward to receiving your revised manuscript.

Kind regards,

Rui Tada, Ph.D.

Academic Editor

PLOS ONE

Journal Requirements:

Reviewers' comments:

Reviewer's Responses to Questions

**Comments to the Author**

1. If the authors have adequately addressed your comments raised in a previous round of review and you feel that this manuscript is now acceptable for publication, you may indicate that here to bypass the “Comments to the Author” section, enter your conflict of interest statement in the “Confidential to Editor” section, and submit your "Accept" recommendation.

Reviewer #2: (No Response)

2. Is the manuscript technically sound, and do the data support the conclusions?

Reviewer #2: Yes

3. Has the statistical analysis been performed appropriately and rigorously? 

Reviewer #2: Yes

4. Have the authors made all data underlying the findings in their manuscript fully available?

Reviewer #2: Yes

5. Is the manuscript presented in an intelligible fashion and written in standard English?

Reviewer #2: Yes

6. Review Comments to the Author

Reviewer #2: The manuscript has improved, yet there are a few areas that require further clarification to enhance the understanding of the experiments conducted and the findings reported:

Comment 7: Please include the results pertaining to the isolation of non-glycosylated proteins. It would be appropriate to add these details in the supplementary data section. Additionally, clarify how you confirmed the identity of the expressed protein as CH2. Were Western blotting techniques used for this purpose? Please elaborate.

Comments 9 and 11: It is necessary to add the reference for the animal protocol that was previously referred to by the authors. This will help in maintaining the integrity of the experimental framework.

Comments 12 and 13: I recommend that you integrate a more detailed explanation regarding these specific concerns within the discussion section of the manuscript. Please ensure that relevant references are cited to support your explanations.

Comment 16: This detail should be included in the methods section of your paper to provide clarity on the procedures used in your study.

Comment 21: The discussion concerning the choice of mouse strain should remain within the discussion section of the manuscript. This will help readers understand the rationale behind the selection and its relevance to the study outcomes.

7. PLOS authors have the option to publish the peer review history of their article (what does this mean?). If published, this will include your full peer review and any attached files.

Reviewer #2: **Yes: **Eakachai Prompetchara

---

## [Author Response · Author response to Decision Letter 1]

29 May 2024

Thank you for submitting your revised manuscript. We appreciate the effort you have put into addressing the comments from Reviewer 2.

Unfortunately, despite multiple invitations and reminders, we did not receive a response from Reviewer 1 during this round of revisions. However, upon careful examination of Reviewer 1's initial comments, we found them to be minor in nature and have determined that you have already adequately addressed them in your revised manuscript.

Considering the thoroughness of your revisions in response to Reviewer 2's comments and the minor nature of Reviewer 1's suggestions, we believe that your manuscript is now suitable for publication after addressing the remaining minor points raised by Reviewer 2.

Please submit your revised manuscript, along with a detailed response letter addressing the comments from Reviewer 2. We look forward to receiving your revised manuscript.

In compliance with the editor's guidelines, we have methodically integrated all feedback provided by the reviewers into the revised manuscript in a systematic, point-by-point fashion as below.

Comment 7

Please include the results pertaining to the isolation of non-glycosylated proteins. It would be appropriate to add these details in the supplementary data section. Additionally, clarify how you confirmed the identity of the expressed protein as CH2. Were Western blotting techniques used for this purpose? Please elaborate.

Response: Thank you for your constructive comments. We have included the purification of CH2 domain in Supplementary Figure 1 and explained why we used non-glycosylated form on page 9, line 176-182. Additionally, we confirmed the identity of the expressed proteins as the CH2 domain using MALDI-TOF-MS measurements, as detailed in our previous study (Oyama et al., JB. 2021) on page 9, line 176-182.

Comments 9 and 11

It is necessary to add the reference for the animal protocol that was previously referred to by the authors. This will help in maintaining the integrity of the experimental framework.

Response: Thank you for your valuable feedback. According to the provided comment, initially, the initial rationale for selecting intraperitoneal injection as the method of administration in the animal protocol of our previous studies by Ohkuri et al., J. Immunol, 2010 [17], Nakamura et al., BBA, 2016 [18], Masuda et al., Front. Immunol., 2022 [21], Oyama et al., BBRC, 2021 [36], and Ohkuri et al., BBR, 2019 [38] followed the route of administration utilized in the pioneering study by Delamarre et al., J. Med. Exp.2006 [19] in the revised manuscript text (page 6, lines 118-122). Moreover, we have explicated the interval immunization protocol currently employed, aligning with the methodology established in prior studies by Ohkuri et al., J. Immunol, 2010 [17], Nakamura et al., BBA, 2016 [18], Masuda et al., Front. Immunol., 2022 [21], Oyama et al., BBRC, 2021 [36], and Ohkuri et al., BBR, 2019 [38] (page 7, line 130-133).

Comments 12 and 13

I recommend that you integrate a more detailed explanation regarding these specific concerns within the discussion section of the manuscript. Please ensure that relevant references are cited to support your explanations.

Response: Based on your comments, we have added a detailed explanation in the discussion section addressing why different protein amounts were used for the adjuvanted and non-adjuvanted groups (to comment 12). It is widely accepted that increasing the antigen dose enhances antibody production, a conclusion supported by our previous studies (Masuda et al., Front. Immunol., 2022 [16]. Therefore, we considered that increasing the antigen dose in the non-adjuvant group is a preferable method for assessing the impact of protein stability on immunogenicity while keeping adjuvant conditions consistent. Additionally, we have added the rationale for using CFA in the first immunization and IFA for the booster in the method section (to comment 13). As previously stated, the content has been integrated as specified, accompanied by proper citation of relevant literature in the Discussion section. (page 12, line 228-241) as well as in the Materials and Methods section（page 7, 126-129) .

---

## [Editor Report · Decision Letter 2]

4 Jul 2024

Relationship between protein conformational stability and its immunogenicity when administering antigens to mice using adjuvants - Analysis employed the CH2 domain in human antibodies -

PONE-D-23-43523R2

Dear Dr. Ueda,

We’re pleased to inform you that your manuscript has been judged scientifically suitable for publication and will be formally accepted for publication once it meets all outstanding technical requirements.

Kind regards,

Rui Tada, Ph.D.

Academic Editor

PLOS ONE
---

## [Editor Report · Acceptance letter]

11 Jul 2024

PONE-D-23-43523R2 

PLOS ONE

Dear Dr. Ueda, 

I'm pleased to inform you that your manuscript has been deemed suitable for publication in PLOS ONE. Congratulations! Your manuscript is now being handed over to our production team.

Kind regards, 

on behalf of

Dr. Rui Tada 

Academic Editor

PLOS ONE